# Paramilitary Conflict in Colombia: A Case Study of Economic Causes of Conflict Recidivism

**William Orlando Prieto Bustos [1,*] and Johanna Manrique-Hernandez [2]**

1   Faculty of Economics and Administrative Sciences, Catholic University of Colombia, Street 47 No. 13-54, Building C, Bogota 110211, Colombia

2   College of Graduate Studies, University of British Columbia, 3333 University Way, Kelowna, BC V1V 1V7, Canada; jemanriq@mail.ubc.ca

*   Correspondence: woprieto@ucatolica.edu.co; Tel.: +57-320-328-3452

**Abstract:** Following the peace accord on 26 September 2016 between the Colombian government and the Colombian Revolutionary Armed Forces (FARC), significant structural issues persisted in Colombia, such as state fragility, land distribution challenges, and rural impoverishment, all of which jeopardized sustainable peace. Previous disarmament events indicated potential shifts in violence and recidivism rates among ex-combatants. This paper aims to determine the likelihood that, in the post-conflict era with FARC, these ex-combatants would rearm themselves into new criminal factions. Employing a methodology by Paul Collier, the study utilized logit, probit, and panel data models with both fixed and random effects to evaluate the recidivism risk at the municipal level. A 1% increase in per capita municipal income decreased conflict probability due to the increased opportunity cost of disrupting economic endeavors. Conversely, 1% increases in potential conflict benefits from tax revenue and natural resource proceeds raised the probability of conflict by 40% and 17%, respectively. Key results indicate that economic advancement, as measured by per capita income, reduced the duration of paramilitary presence, whereas revenue from taxes and natural resources extended it at the municipal level in Colombia.

**Keywords:** post-conflict; criminal gangs; peace negotiations; urban and rural economy

## 1. Introduction

From 1960 to 1980, there were at least sixteen civil conflicts around the world that achieved peace accords. In Latin America, the civil conflicts in El Salvador, Guatemala, Haiti, Peru, and Colombia reached peaceful agreements in 1992, 1996, 2004, 2000, and 2013, respectively. In Africa, specifically in Liberia, Mozambique, Rwanda, Sierra Leone, and South Africa, they made possible nonviolent conflict settlements in 2003, 1992, 1994, 2002, and 1991. Moreover, Afghanistan, Armenia, Cambodia, and Timor-Leste in Asia achieved a full stop of violence in 2001, 1994, 1990, and 2002. Furthermore, the Bosnia and Herzegovina and Kosovo wars in Europe achieved peace accords in 1995 and 1999, respectively (Prieto-Bustos and Manrique 2018).

Previous experiences have shown that there are political, economic, cultural, and social factors that could increase the likelihood of conflict recidivism when there is not enough time and insufficient resources to provide a stable environment in which there is the consolidation of a system of violence, disputed economic resources among illegal armed groups, lack of trust and participation in the post-conflict institutional settlement, and absence of a community wounds-oriented justice system.

In this regard, post-conflict research is relevant as it provides systematic evidence regarding environmental conditions related to reducing violence resurgence as a mechanism to resolve social conflicts. Moreover, post-conflict research provides the opportunity to build a theoretical paradigm of conflicts that overcomes the singularities and the comparative conflict analysis, revealing a theoretical hypothesis that could be useful in researching

conflicts worldwide. Although each conflict has different causes related to the historical and contextual aspects of power relations that frame its social, economic, cultural, and political development, going from the particular aspects to general aspects is a relevant theoretical effort as long as violence as a resource for resolving social conflicts remains a human experience.

The Colombian post-conflict is relevant in such an endeavor for at least five reasons. First, the Colombian conflict is among the oldest conflicts on the planet, beginning in 1964 and ending with a peace accord in 2013. In this regard, almost 12.2 million people lived in rural areas, and more than 9 million were displaced, from which almost 5 million were women and 2.1 million were young people. At the beginning of the decade of the 1960s in Colombia, the armed confrontation of political dissidence between the 1940s and 1950s enabled the emergence of the Revolutionary Armed Forces of Colombia (FARC). After more than 50 years of violence, the Colombian government has reached a deal with that organization to end the conflict. Nevertheless, the end of the violence goes beyond the peace accord, as Colombia has deep structural problems of economic, social, and geopolitical order, which can lead the violence not to cease but to transform into another kind. That was the case in the negotiated agreement with leaders of the United Auto-defenses of Colombia (AUC), an organization of a paramilitary kind, which made its first appearance during the decade of the 1980s and reached its pinnacle in the 1990s. Fears about the problem of recidivism of paramilitaries problem shades the results of what would happen in the case of the peace process with the FARC guerrillas.

Fear about the recidivism of paramilitaries problem shades results in what would happen in the case of the peace process with the FARC guerrillas.

Second, until recently, due to the revelations of the Truth Commission and the transitional justice system, the Colombian civil conflict was accepted by Colombian society as a civil conflict. Although, in 2011, the Supreme Court of Justice and the Colombian government issued sentences and laws to protect and reinstall the victims' human rights against violations, in 2016, a political plebiscite among citizens implemented to approve or disapprove of the peace accord showed a negative response against the 2013 peaceful agreement based on the argument that guerrillas were terrorist groups. Hence, the human rights conflict approach was only partially accepted.

Third, right-wing extremist groups' response to left-wing guerrillas was to create paramilitary armies that, in some cases, work hand in hand with the national army, leaving the government's monopoly power of violence as a resource for ring-wing extremist groups. Fourth, there has been a historical dispute over territories that have natural resources with high international prices, such as cocaine plants, gold, and emeralds, among others, and geographical advantages in producing and exporting such natural resources. Fifth, there are social and economic gaps between citizens who live in metropolitan cities and citizens who live in rural areas due to a lack of strategic physical and social infrastructure investment, absence of a development strategy for rural areas and regions within rural areas, corruption and coaptation within local governmental institutions, and lack of an optimal taxing strategy to create economic incentives for better use of natural resources in rural areas, among others.

Furthermore, poverty reductions are not associated with better income distribution in Colombia. Although economic growth has been increasing on average by 3%, there have yet to be relevant improvements in institutional quality that provide better opportunities for income distribution. The relevant changes created by the 1991 new constitution regarding citizens' participation mechanisms through local democratic elections, social manifestation rights, and access to justice have not yet guaranteed inclusion in an economic model that focuses on economic development in rural areas. For example, in rural areas, there needs to be more cooperation between labor markets and educational access in local and regional environments without proper strategic investment in physical, social, and technological infrastructure. Therefore, social participation has created a demand for education, but there is a strategic development plan that connects access to education with employment

or self-employment opportunities. Thus, social participation has reduced poverty without significant changes in income distribution, creating a potential for conflict recidivism.

At the macro and meso-institutional levels, there has been progress toward independence and power allocation to the judicial, legislative, and executive government branches. However, there have still been few advances in achieving optimal use of natural resource royalties managed by the National System of Royalties (NSR) and the social investment distribution system of the central government, frequently used to pay political favors. Moreover, land property rights and suboptimal economic incentives have delayed the development of the land market due to transactional costs and the need for an optimal tax policy. Therefore, there are few possibilities for income generation for citizens in rural areas who were displaced, including access to goods and services relevant to avoid participation in criminal markets. In addition, the lack of government presence in rural areas has created opportunities for the consolidation of a violent system created by illegal organizations that coopt and corrupt local institutions, use violence to control territories, influence democratic elections, and destroy social cohesion by taking away young people into criminal activities (Giraldo et al. 2015). Also, the displacement caused by the violent system in rural areas has added pressure on environmental resources and social practices, leading to discrimination and exclusion in urban areas (Prieto-Bustos et al. 2021) and deforestation and predation of natural resources. In this regard, the post-conflict requires a significant rural development policy (Ocampo 2015) to reduce the likelihood of recidivism.

This paper aims to build a probabilistic model to identify regional risk factors associated with a higher likelihood of recidivism of violence in Colombia following the signing of the peace agreement in 2016. To achieve this, a macroeconomic model employing spatial econometrics refines microeconomic data to ascertain the probability of a resurgence in violence at the regional level in Cundinamarca, Colombia.

According to previous studies, to reduce the probability of relapse of a conflict, it is mandatory to strengthen institutions, promote socio-economic development, and invest in social capital. Boosting local productive capacity increases the chance of former conflict actors entering the formal labor market to guarantee their income stability and reduce the possibility of them returning to crime. Nevertheless, credible political commitments are crucial to aiming for lasting peace.

This research illuminates the issue by modeling the effects on the likelihood of violence at a regional level. It augments previous studies in terms of theoretical and empirical contributions by conceptualizing informal markets as violence markets and introducing a new graphical index map of regional risk. This map is employed to estimate the spatial interrelation of factors linked with a higher likelihood of experiencing violence in Colombia, using a risk index. The paper is divided into four main sections in addition to the introduction. The second section surveys the state of the art in conflict studies, discussing both the international and national scientific literature on conflict recidivism. The third section delves into the geography of Colombia's paramilitary conflict. The fourth section outlines the estimation strategy, research hypothesis, methods, data, and the results obtained. The concluding section offers the research's key findings.

## 2. Literature Review

### 2.1. International Literature

The institutional development literature emphasizes the mechanisms through which international donors became involved in reconstructing post-conflict societies. While first-generation studies of international cooperation aid generally showed persistent interventions aimed at economic growth, inclusion, and human development, recent evidence has revealed a heightened sensitivity to institutional contexts in which specific governance models might have been more suitable. In this vein, the second generation highlighted endogenous aspects of building trust, consensus, and social policy for those most affected by conflict, rather than focusing solely on tax-based reconstruction reforms. Conversely, mechanisms related to disarmament, demobilization, reintegration, and addressing the

psychosocial trauma of conflict were crucial prerequisites for minimizing the chances of armed conflict resurgence.

Tobias and Boudreaux (2011), Englebert and Tull (2008), Collier (2009), Rugumamu and Gbla (2003, 2004), and Kreimer (2000b) provided examples of institutional developments in post-conflict African societies. These were primarily marked by efforts to achieve (1) economic growth, (2) capital accumulation, and (3) security (namely, conditions of disarmament and reintegration). Key among the economic development strategies was the promotion of cooperative economic models for coffee cultivation. This was bolstered by the strengthening of small enterprises, fostering innovation in urban agriculture, and formulating economic policies that drove growth but also emphasized redistribution to populations most impacted by the armed conflict. For instance, Tobias and Boudreaux (2011) explored the pivotal role of entrepreneurship in poverty alleviation, economic development, conflict mitigation, and broader social progress.

Furthermore, Lemmon (2012) underscores the positive economic impacts of facilitating credit access under conditions that promote corporate sustainability. Key requirements include support for the development of viable investment plans, market access, connectivity to corporate networks, and training in essential business skills like record-keeping, product marketing, and sales techniques. Without these fundamental elements, an employment and income-generating strategy based on microenterprises might falter, even with access to credit, due to intense market competition and more sophisticated business environments. However, with these foundational elements in place, such a strategy can succeed, especially when bolstered by credit access in competitive and well-prepared settings.

On the other hand, Lynch et al. (2013) conducted research in Sierra Leone that described the conditions of the displacement of agricultural labor to the cities as a consequence of armed conflict, highlighting it as an opportunity for developing urban agriculture. The study confirmed the importance of urban agriculture for food security, employment generation, and the reconstruction of displaced communities in urban enclaves of agricultural production.

Meanwhile, in an economic reform approach, Ansoms (2005) highlighted the priority redistribution approach to conflict victims as part of reforms seeking to stimulate economic growth. In this regard, that study proposed a strategy that combined growth and wealth distribution to effectively reduce poverty in post-conflict conditions.

Moreover, issues related to human development emphasized the importance of fairness and equality in the labor market's functioning in post-conflict societies. Leadership as a strategy for social reconstruction, the relevance of the human rights approach, the gender perspective in post-conflict social policy, and the violent consequences for children exposed to conflict were relevant factors in achieving sustainable peace. In particular, Smet (2009) described the case of Sierra Leone concerning sustainability and the promotion of gender equality by identifying the impact of the armed conflict on gender roles. The study provided evidence of the need for legislative changes aimed at abolishing discriminatory practices and internalizing within the community the importance of education and women's participation in the labor force, for both the economic and social reconstruction of communities.

Furthermore, Akresh et al. (2011) explored the correlation between the destruction of crops during conflict and its harmful effects on the health and education of children who experienced armed conflict. This correlation resulted in unfavorable social and economic outcomes in terms of the biological and cognitive development of infants. Exposure to war reduced the height and weight of infants at the time, affecting the human development of future generations to the extent that it decreased the potential for accumulating relevant capacities for exploiting opportunities in post-conflict Rwanda. In addition to the relevance of reforms designed to eliminate discriminatory behavior and the harmful effects on the human development of a generation of infants exposed to conflict, UNDP (2010) emphasized the priority of leadership as a strategy to boost confidence, legitimacy, and social transformation in post-conflict societies.

Leadership plays a pivotal role during transitional periods, especially in aligning and motivating change. It fosters social cohesion and reconnects society with the economic and social reconstruction that defines post-conflict environments. Massaquoi (2007) delves into the human rights approach in relation to the female population in Libya, viewing it as a strategic method for rebuilding and strengthening state institutions post-conflict. For example, leadership rooted in a women's rights perspective is vital for the rejuvenation of institutions and restoring public influence over social policies. Patel (2005) highlights successful instances in Asia where community empowerment anchored in gender equality yields greater social benefits, particularly in terms of job creation in Afghanistan. Similarly, Iyer and Santos (2012) document increased job opportunities within post-conflict Asian societies, especially through public investment initiatives that reintegrate unskilled workers into the labor market.

Kreimer (2000a) describes how institutional structures streamline the management of resources provided by international donors. Particularly, the coordination of activities during the design, financing, and implementation stages of Bosnia–Herzegovina's reconstruction was crucial when paired with resources from international donors, as emphasized by Ohanyan (2002). Furthermore, Ding and Wallich (1996) note that community involvement in project design, rooted in institutional structures for financial management, is pivotal for maximizing social benefits during the rebuilding of local governments. This approach enhances planning, decision-making, and mobilization in the post-conflict era. Haynes (2010) stresses the importance of leadership in project implementation. Participatory dimensions prove most effective when a human rights approach is used, especially when targeting the development of the female population. Such an approach ensures adherence to integrity and dignity principles vital for successful social reconstruction projects. Huerou (2010) further underscores citizen participation as essential for the long-term sustainability of peace processes, leading to increased returns on investments aimed at bolstering local governments' capabilities.

Garstka (2010) elaborates on the participatory principle in project design, highlighting a project related to the formalization of illegal neighborhoods in Kosovo. This project combined resources from international donors, the United Nations' methodological expertise, and local public participation, presenting a viable solution for job creation and social reconstruction. McMahon (2004) identifies three key pillars of international support in Bosnia–Herzegovina: (1) fostering sustainable agreements between parties, (2) enhancing governmental structures, and (3) rejuvenating and sustaining the financial system. Klapper and Panos (2009) emphasize the importance of financial systems as vital institutional setups for rebuilding and transitioning to sustainable entrepreneurship.

In the context of Latin American post-conflict experiences in El Salvador, Guatemala, and Colombia, it is worth noting that these conflicts were rooted more in economic grievances than ideological divides. This has resulted in a greater emphasis on income and job generation in Latin American post-conflict scenarios, as opposed to participatory strategies. Eriksson et al. (2000) highlight the institutional arrangements that fostered macroeconomic stability, which was instrumental in El Salvador's post-conflict economic boom. Such institutional reforms are vital as they enhance the credibility of international donors. Tabak (2011) observes that flaws in the judicial system, especially concerning gender rights in transitional justice measures, can hinder reconstruction efforts. Thus, while macroeconomic reforms spur economic growth, robust transitional justice systems strengthen institutional foundations for lasting peace. The predominant challenges in Latin American post-conflict scenarios largely revolve around the micro-level operations of justice systems.

Lastly, the ILO (2010) underscores the importance of local economic recovery (LER) in post-conflict settings. The LER strategy hinges on job creation, prioritizing economic progression over local institutional reforms. Adhering to the guidelines for decent work, the ILO proposes five methodologies, encompassing (1) employment and labor market

assessment, (2) value chain analysis, (3) consumer surveys, (4) vulnerability assessments, and (5) training needs analysis.

Despite the significance of employment, it is crucial to acknowledge the prevalent high concentration of land ownership in Latin American countries. This concentration often serves as a structural catalyst for economic grievances that spark violence, notably in Colombia. For example, Thomson (2011) underscores land ownership as a factor that destabilizes the Colombian peace process. Rooted in the evolution of capitalism, agrarian disputes epitomize the structural basis for violence in Colombia. Stemming from this, the dominant role of capital in the prevailing economic model obstructs the adoption of community property schemes better suited for local post-conflict economic development. Consequently, as Thorsell (2013) also suggests, promoting entrepreneurship within a neoliberal framework may be ineffective in generating income and ensuring economic stability in economies with pronounced wealth disparities.

Several studies examine the implications of violence and poverty on gender dynamics. The migration induced by economic motives and the forced displacement due to conflict result in profound shifts in the societal contexts where culturally defined gender roles play out. Menjivar and Agadjanian (2007) delve into how changes in gender dynamics, precipitated by such displacements, lead to marked economic and cultural shifts. By juxtaposing Guatemala and Armenia, their research illuminates the societal changes where women assume roles as primary income earners, replacing men.

Additionally, from a methodological standpoint, contrasting diverse post-conflict experiences is invaluable. For instance, Nasi and Rettberg (2005) note the unique facets of the Colombian conflict, which historically justified its omission from comparative post-conflict studies in Latin America. Nevertheless, more recent research that incorporates Colombia has enriched the realm of theoretical and empirical inquiry, shedding light on the nation's violent history. Randall (2005) advocates for a shift from studying conflict demographics to exploring the demography of conflicts, unveiling intricate relationships rooted in economic and ethnic disparities. Such an approach makes the structural triggers of armed conflicts more discernible across varied post-conflict societies.

Emphasizing the inherent nature of conflict further accentuates the endogenous character of post-conflict recovery. This perspective elevates the importance of participatory approaches. Lawrence (2007) posits that, given the internal essence of post-conflict recovery, indigenous factors should be prioritized over external ones as violence catalysts. Consequently, participatory initiatives centered on nurturing social capital and trust become paramount for achieving lasting peace. Absence of community involvement in post-conflict strategies might let radical political stances erode initial positive gains. Braithwaite and D'Costa (2015) illustrate how such extremist political positions in post-conflict societies beget violence. Furthermore, heavy-handed tactics to reassert control in areas devoid of state presence can instigate a cyclical pattern of state-sanctioned violence and war crimes. Hence, security strategies emphasizing community involvement are essential for sustaining peace. Onoda (2004) discusses the role of security pacts as vital institutional tools for ensuring long-term peace.

The concept of power elites offers insight into this issue. As originally posited by C. Wright Mills (2018), a select group from the political, corporate, and military sectors wields significant influence over national affairs. Mills' observations highlight the dynamics of how and why local conflicts can be influenced or organized by this group, especially those of a paramilitary nature.

Subsequent authors expanded this concept. Domhoff (1978) delved into the relationship between economic elites and political decision makers, suggesting that power is concentrated within a few families with a large influence, who decree national policies, sometimes encouraging local paramilitary forces to safeguard their welfare.

In Jeffrey Winters' exploration of power dynamics (Winters 2011), he posits that wealthy individuals and groups protect their assets and influence by subtly shaping policies and sparking conflicts that distract from their actions. Within this context, actors who might

appear less visible or less powerful can still have a significant impact on national politics and decisions.

Likewise, Sheldon Wolin (2017) explores this idea but more in relation to corporate powers, who would dominate traditional democratic institutions, leading to potential localized conflicts because of larger national and international strategies. Wolin's work implies that local conflicts can be seen not just as isolated incidents but as manifestations of a broader power structure influenced by elite interests. This is called 'inverted totalitarianism' and describes a system in which corporations discreetly control and manipulate democratic processes without overt leadership. In this paradigm, corporate interests silently and anonymously co-opt democratic structures. A continuous state of warfare and emergency further consolidates power.

Nancy Fraser (2017) and Saskia Sassen (1991) also examine the relationship between national elites and local paramilitary conflicts, focusing on neoliberalism and how global capital flows and elite decisions can affect local communities, sometimes leading to conflicts. Sassen (1991) explores the concept of global cities where local conflicts and paramilitary groups may find support from hidden ties with power elites.

From the international literature review, several converging themes from the examined post-conflict experiences emerged. First, historical conditions played a pivotal role in defining the demographics of conflict. Second, post-conflict recovery was an endogenous process that proved more stable when a participatory approach was adopted. Third, peace accords necessitated binding conditions grounded in community engagement, which needed regular monitoring and evaluation. In addition, targeted initiatives in institutional fortification, local capability enhancement, societal and personal safety frameworks, revitalization of financial systems, and investment in human capital have been vital in fostering sustainable peace. The subsequent section succinctly outlines the historical backdrop of the Colombian civil conflict to pinpoint factors linked with a heightened likelihood of conflict recidivism.

*2.2. Social Dynamics and Institutional Context of the Colombian Civil Conflict*

In the early stages of state-building after Colombia's independence, conservative forces sought to prevail over the liberals. This political class struggle transcended several dimensions: political, economic, and social. Such battles shaped the prevailing perception that elites dominated the government, establishing it as a closed system marked by a persistent discourse of exclusion toward the less affluent. In this context, the relative strength or weakness of the Colombian state has remained a central topic in discussions about the origins and persistence of violence.

The initial stages of configuring the Republic lacked cohesion, resulting in the institutional framework being primarily developed in urban centers rather than in the suburbs or rural areas. In these outer areas, large landlords, known as "gamonales", maintained territorial control, upholding feudal institutional figures that had roots in the preceding colonial era. Daniel Pecaut (1997) was among the early authors to elucidate the connection between the political regime, state-building, and violence. The state's fragility prevented it from expanding, establishing market rules, and reigning in local political powers. This dynamic between the state and society could lead to social tensions. This was the core premise of Migdal et al. (2011), who subsequently examined violence, social groups, and the rise of social and political conflicts. Additionally, Cubides et al. (1998) detailed how the multifaceted nature of violence comprised numerous violent acts; hence, a failure to recognize this complexity made it challenging to pinpoint the origins of violence and its association with state-building.

Conversely, Paul Oquist (1978) stressed that the state's presence incited civil conflict. During its formation, the two dominant parties, driven by their aspirations for dominance, established uniform patronage networks which did not truly represent the populace and offered no room for negotiation. As a result, numerous social disputes from the nineteenth century remained unresolved, coexisting with episodes of violence and civil wars. This overwhelmed the state's capacity to quell civil discord and impaired its responsiveness

to the people's needs. Similarly, Reyes Posada and Duica Amaya (2009) spotlighted the government's incapacity to address these predominantly agricultural social and territorial disputes, weakening the state's operations and eroding its monopoly on order.

The Commission on the Study of Violence (Jaramillo 2011) pointed out a pronounced vulnerability of the state due to a deficit in democratic practices, especially locally. Consequently, fostering social inclusivity became paramount for reinforcing a democratic ethos that genuinely engaged the citizenry. The 1991 constitution, superseding the one from 1886, was championed by social factions that emphasized a participatory agenda aimed at achieving inclusivity.

In the same way, Uricoechea (1986) contended that the state was not weak; its magnitude in terms of funding and staff for various public agencies, along with social investment, had grown in line with the increasing demands and responses to the needs of the vulnerable population. However, it was not representative of social demands because of a limited democratic inclusion of social forces at a local level due to the poor quality of democratic processes in local and rural areas.

Furthermore, González (2009) refers to such weakness in political representation and strength in government growth as normative categories that do not correspond to state-building but have created overlapping capabilities within the state. Lack of representation leads to state absence and growth in the size of government, implying that state presence has yet to be met at rural local levels. However, a nation should be an articulated social project built from the territories so that their fragmented capacities (González et al. 2012) to participate in and assemble the state is a conflict driver that goes beyond the argument of either a present or absent government.

In addition to the weakening of the state, the political system is one of the most important reasons related to the emergence of social conflicts that led to the formation of guerrilla groups in the mid-twentieth century. A lack of democracy, polarization between two political parties, and clientelistic structures that pervaded political positions weakened state control over the masses, particularly those outside the centralized institutional order. Moreover, according to Sanchez and Peñaranda (1986), the ruling elites of both political factions actively encouraged armed confrontations among the lower classes, a phenomenon they termed "endemic warfare".

In general, four main points converged from the reviewed literature: the fragility of the state in the face of restrictive democracy, the absence of regulatory boundaries, particularly for local authorities, the link between institutional bottlenecks and violence, and the political regime's association with local violence. Pecaut (1997) believed the issue stemmed from a state system that was not politically rigid but somewhat adaptable, leading to challenges in structuring and moderating local interests. Conversely, Carbo (2006) argued that bipartisanship resulted in social exclusion and catered exclusively to the elite, thereby generating societal tensions. Consequently, the institutional frameworks that underpinned the state system indicated a plausible link between the state and violence. Vasquez (2014) and Garcia de la Torre (2004) contended that the construction of social identity within a territorial space shaped violence when governance lacked inclusivity. Political decentralization yielded mixed outcomes for state-building processes. While Ritter Gutierrez (2022) illustrated how political decentralization was vital for state-building at the grassroots, Fals-Borda (1991) proposed that meaningful decentralization should center on the interplay of social movements, identity, and violence within a territory. From this, regional institutional analysis should emphasize two dominant themes: (1) the role of political decentralization in molding the state's relationship with territorial social identities and (2) institutional shortcomings that incite regional violence.

Political decentralization, municipal advancement, and unlawful economic proceeds closely correlated with aggressive local developments (Sanin et al. 2006), augmenting the danger of government co-optation for private gains. Drug trafficking and paramilitary activities, for instance, were linked with spatial elements like drug routes and land ownership. Lopez Muñoz (2019) segmented the paramilitary phenomenon into three phases:

(1) the rise of paramilitary forces, (2) their autonomy, and (3) their local dimension. Initially, the paramilitary surge was a response by local elites and the state to insurgent movements. However, the allure of the narcotics trade's international revenue provided an economic rationale for exerting social control over territories to penetrate the global illicit drug market.

Subsequently, paramilitary forces gained autonomy, becoming less aligned with the state's ideology (Medina Gallego 1990). They resorted to forced displacement to confiscate rural lands, enhancing political susceptibility to local elites and degrading democratic processes. The paramilitary forces' violent tactics redefined local dynamics. Reyes Posada and Duica Amaya (2009) and Medina Gallego and Tellez Ardila (1994) recounted the state's diminished control over the paramilitary movement and the latter's eventual domination of territories, evolving into a multifaceted paramilitary initiative, from state terrorism to regional crime syndicates.

Interactions between political entities and illicit groups controlling territories unveiled shared interests in unlawful activities (Gutierrez-Sanin 2022). Meanwhile, the paramilitary movement infiltrated and co-opted local governance, defying the central administration, a phenomenon evident in the parapolitics scandal (Romero 2003). González et al. (2002) believed such dynamics fostered 'armed amorality', mirroring the objectives of local elites and agribusinesses within paramilitary factions. Though the inception of paramilitary forces was a countermeasure against guerrilla movements (Romero and Torres 2011), both entities discerned illegal revenues as pivotal to their economic and political missions. This led to a fluid social transformation where the local institutional landscape was revamped by the decentralization process, enabling violent control mechanisms by the paramilitary to function within the state's confines (Garay 2009; Lopez Muñoz 2019). Consequently, drug trafficking, illicit mining, extorting multinational corporations, proprietary financial assets, and corrupting local public institutions became the primary revenue streams for the paramilitary initiative (Richani 2010; Pearce 2007).

To comprehend these funding avenues, Duncan (2005) posited that they offer insight into the paramilitary's evolution into a criminal syndicate amidst the Colombian civil unrest. Likewise, Romero (2003) championed a profound grasp of economic dividends when detailing the paramilitary strategy. The preeminence of economic motivations over ideological factors in both the paramilitary and guerrilla agendas is not novel. Rangel Suarez (1998) contested that structural and ideological shifts were not the principal drivers behind territorially based violent endeavors. For example, illicit profits augmented guerrilla revenues to approximately 1000 million pesos in 1998, primarily sourced from drug trafficking and extortion. This narrative solidified during the 1990s when guerrilla factions exerted dominance in regions including Cauca, Putumayo, and Guaviare, levying taxes on locals through coercion and influencing electoral outcomes. In the paramilitary's case, illegal revenues funded training and weaponry procurement, equipping them with formidable arsenals.

### 3. Colombian's Geographical Paramilitary Conflict

Paramilitarism in Colombia first emerged in the 1950s when many landlords began forming their own armies. However, it was not until the 1980s that it expanded in terms of operations, combatants, and geographical presence. At the departmental level, the extent of the paramilitary phenomenon is gauged by the years since their first act of violence. There is also a consideration of the recidivism observed in each department, showing patterns of re-organization after negotiations with the Colombian state. Table 1 describes paramilitary violence at a departmental level regarding dates of massacres.

**Table 1.** Summary of the most important dates of the events of massacres perpetrated by paramilitary groups at departmental level.

| Department | Start Year | Last Year of Significant Reduction | Date | Recidivism | Last Date |
|---|---|---|---|---|---|
| Antioquia | 1982 | 2003 | 26-December-03 | 2004–2012 | 07-November-12 |
| Arauca | 1998 | 2004 | 31-December-04 | | |
| Atlantico | 1994 | 2003 | 21-May-03 | | |
| Bolivar | 1991 | 2003 | 16-March-03 | | |
| Boyaca | 1982 | 2004 | 16-December-04 | | |
| Caldas | 1990 | 2003 | 08-June-03 | 2005, 2010 | 09-January-10 |
| Caqueta | 1980 | 2001 | 15-February-01 | 2002, 2003, 2005, 2006, 2007, 2011 | 26-November-11 |
| Casanare | 1988 | 2000 | 17-October-00 | 2005, 2009 | 29-April-09 |
| Cauca | 1991 | 2002 | 08-November-02 | 2011 | 20-April-11 |
| Cesar | 1983 | 2002 | 08-December-02 | 2003–2005 | 04-December-05 |
| Choco | 1990 | 2002 | 17-April-02 | 2007, 2008 | 03-May-08 |
| Cmarca | 1990 | 2005 | 04-September-05 | | |
| Bogota | 1989 | 2003 | 08-March-03 | 2009 | 01-December-09 |
| Cordoba | 1988 | 2003 | 06-May-03 | 2008, 2010, 2011 | 25-September-11 |
| Guaviare | 1986 | 2004 | 16-September-04 | | |
| Huila | 1988 | 2006 | 15-May-06 | | |
| La Guajira | 1992 | 2005 | 08-May-05 | | |
| Magdalena | 1991 | 2003 | 16-October-03 | | |
| Meta | 1988 | 2002 | 13-October-02 | 2003, 2004, 2006, 2012 | 14-February-12 |
| Nariño | 1999 | 2003 | 26-April-03 | 2007–2011 | 25-June-11 |
| Norte Sder | 1989 | 2004 | 25-December-04 | 2010, 2011 | 18-June-11 |
| Putumayo | 1991 | 2005 | 27-July-05 | 2011 | 11-February-11 |
| Risaralda | 1992 | 2003 | 26-February-03 | | |
| Santander | 1982 | 2003 | 07-April-03 | 2005, 2009, 2010 | 15-Janaury-10 |
| Sucre | 1992 | 2003 | 12-August-03 | | |
| Tolima | 1989 | 2005 | 15-September-05 | | |
| Valle Cauca | 1986 | 2005 | 09-July-05 | 2006, 2011, 2012 | 20-March-06 and 18-October-12 |
| Vichada | 1998 | 1999 | 20-May-99 | | |

Source: Own calculations based on the Centro de Memoria Historica.

At the national level, the paramilitary conflict had begun by 1980, marked by the first actions of paramilitary groups. Specifically, a massacre was reported in the municipality of Puerto Rico, in the department of Caqueta. On the departmental scale, the earliest paramilitary forces were found in Caqueta, Antioquia, Cesar, and Boyaca. These departments were notable for illegal mining activities (Boyaca and Antioquia) and narcotraffic production (Caqueta), and they provided routes to international markets (Antioquia and Cesar). Casanare, Guaviare, and Meta, known for their oil deposits and access to the natural and mineral resources of the Amazon jungle, had witnessed the presence of paramilitary forces since the late 1980s. During the first half of the 1990s, La Guajira, Magdalena, Bolivar, Atlantico, Choco, and Cauca saw a surge in paramilitary activities, coinciding with the growth of illegal narcotraffic income due to the control of export routes along both the Atlantic and Pacific coastlines. Between 1995 and 2000, paramilitary presence was evident in Vichada and Nariño, and in the capital city, Bogota, paramilitaries had been active since 1989. In 15 out of 28 departments, there were repeated instances of paramilitary activity; notably, Caqueta and Meta experienced five and four recurrences, respectively. Such patterns underscored the challenges in curbing and managing paramilitary forces, especially in departments rich in natural resources and prone to narcotraffic revenues.

Figure 1a,b refer to paramilitary violence within Colombia, characterized by massacres and conflict duration graphs. Illustration 1 shows that most of the territories suffered from at least one event related to paramilitary presence. The Amazon and Orinoquia regions in the south and east of Colombia were not significantly affected, as guerrillas have

traditionally dominated these areas. Additionally, Illustration 2 displays the territories with the longest durations of paramilitary conflict.

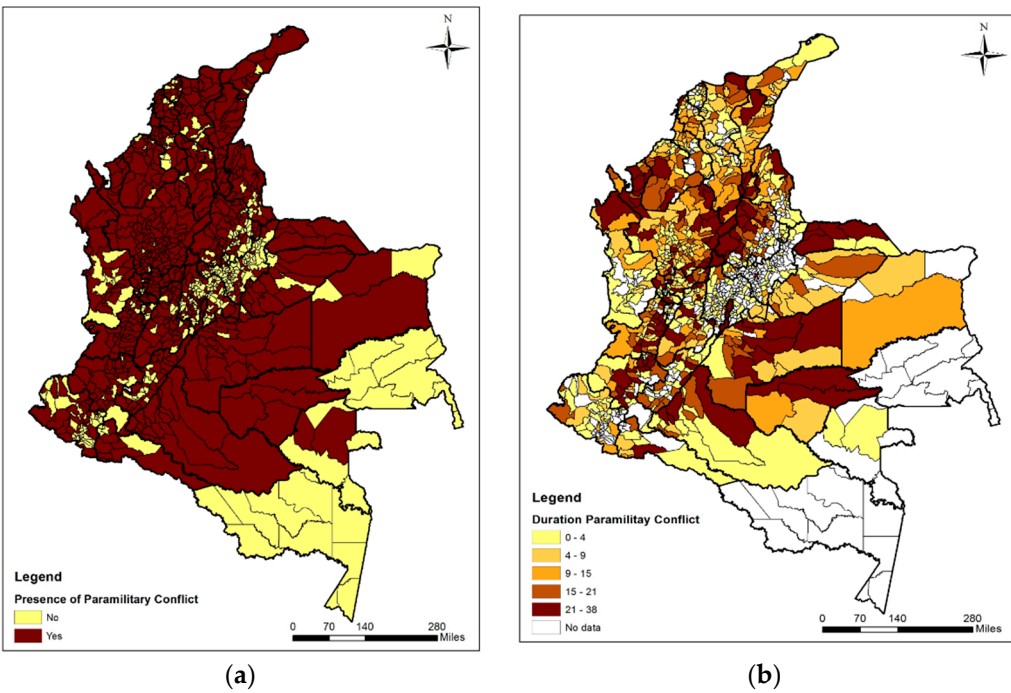

| (a) | (b) |

**Figure 1.** (**a**) Presence of paramilitary conflict; (**b**) length of paramilitary conflict.

Furthermore, the most substantial rise in paramilitary massacres during the conflict took place between 1996 and 2003, accounting for 5057 victims. This period alone represented 69.58% of all victims from such events in the paramilitary conflict up to 2012. At the department level, Antioquia bore the brunt of paramilitary violence during this observed period, suffering 2232 murders in massacres between 1980 and 2012. It was followed by Santander (526), Cesar (494), Bolivar (456), Norte de Santander (455), Magdalena (399), Valle del Cauca (386), Meta (325), Cauca (275), and Cordoba (253). Antioquia alone accounted for 30.74% of the victims of massacres committed by paramilitary groups in that period, along with Santander (7.26%), Cesar (6.8%), Bolivar (6.3%), Norte de Santander (6.9%), Magdalena (5.5%), Valle del Cauca (5.3%), Meta (4.5%), Cauca (3.8%), and Cordoba (3.5%). Those ten departments encompassed 79.71% of the massacre victims recorded between 1980 and 2012.

Grouping departments into regions illustrated that the central west region of Colombia was the most impacted by paramilitary violence between 1980 and 2012. The departments of Antioquia, Risaralda, Caldas, and Quindio belong to the central west region. In contrast, Norte de Santander, Santander, Boyaca, Cundinamarca, Bogota, Tolima, and Huila comprise the central east region. The departments of Atlantico, Bolivar, Cesar, Cordoba, Guajira, Magdalena, and Sucre constitute the Atlantic region. Valle del Cauca, Choco, Cauca, and Nariño form the Pacific region, while Orinoquia, Amazonia, and all other departments fall into the New Departments region. The central west region saw the most conflict, with a total of 2338 massacres (32%), trailed by the Atlantic region with 1975 massacres (27%), central east region with 1433 massacres (20%), Pacific region with 881 massacres (12%), and the New Departments region with 632 massacres (9%).

## 4. Econometric Model: Do Civil Wars Have Economic Causes?

The econometric model proposed is based on Collier (2009). According to Collier's findings, a country faces two primary challenges when forging a durable peace agreement: (1) catalyzing economic recovery from the severe repercussions of war and (2) diminishing the risk of conflict relapse. Collier provided evidence that half of the civil wars experienced

in the 1990s resulted from a return to warfare following a peace agreement. In many instances, conflict recurrence occurred within the first decade following the peace accord. Due to limited data, modeling this was challenging until similarities in certain African countries were taken into account. Collier began by identifying specific factors associated with the onset of conflicts, such as low per capita income, the state's inability to assert control over its territory, sluggish economic growth, and a high dependency on the export of economic resources.

Using a utility-theory-based model, Collier posited that civil war is contingent upon the benefits outweighing the costs of rebellion. Employing tobit, probit, and duration models, Collier et al. (2008) sought to ascertain the likelihood of civil war and its duration based on four variables: initial income, ethnolinguistic fractionalization, the abundance of natural resources, and initial population size. The onset of conflict hinged on the probability of victory, and the factors influencing this probability included the government's defensive capabilities, military expenditure, financial resources allocated to national defense, and tax rates or the per capita taxable base. A higher likelihood of a rebel victory was inversely related to the state's capacity, as gauged by its military strength and per capita tax base. Furthermore, Collier's model indicated that the propensity for relapse was influenced by the benefits and costs tied to the conflict. Gains were derived from seizing state power, while costs entailed opportunity losses stemming from hampered economic activity and transaction costs arising from coordination efforts within the insurgent group. Table 2 outlines both the benefits and costs of conflict recurrence.

**Table 2.** Gains and costs of conflict relapsing.

| Gains | | Costs | |
|---|---|---|---|
| **Explanatory Variables** | **Proxy** | **Explanatory Variables** | **Proxy** |
| Per capita taxable capacity of the economy | Per capita income and natural resource endowment | Loss in income due to the conflict | Per capita income |
| Benefits of secession | Size of population | Cultural distinctness | Ethnolinguistic fractionalization |

Source: Collier et al. (2008).

Using Collier's approach, Equations (1) and (2) describe the economic model according to the adjusted panel data about the paramilitary violence in Colombia. Equation (1) describes the probability of conflict relapse, while Equation (2) describes the determinants of the number of years with the presence of paramilitary organizations. Both equations are adjusted at the municipality level. The model's purpose is to contrast re-search hypotheses against available data to show whether the economic aspects are relevant for explaining the occurrence of conflict relapse in Colombia.

$$\text{Conflict}_{it} = \beta_{0it} + \beta_1 \text{PMI}_{i,t} + \beta_2 \text{PTI}_{i,t} + \beta_3 \left(\frac{\text{NRR}}{\text{TMI}}\right)_{i,t} + \beta_4 \text{MA}_{i,t} + \beta_5$$
$$+ \left(\frac{\text{MI}}{\text{TMI}}\right)_{i,t} + \beta_6 \text{RPD}_{i,t} + C_i + D_i + U_{it} \tag{1}$$

Conflict is a variable that takes 1 when the municipality "i" exhibits a violent paramilitary event in a year "t" and 0 when it does not. PMI is the per capita municipality income for the municipality "i" in year "t". PMI equals total municipality income divided by the total population. PTI is the per capita tax income for each municipality in year "t". NRR represents natural resources royalties the municipality "i" received in the year "t". TMI is the total municipality income for each municipality "i" in the year "t". The proportion of NRR divided by TMI is an instrument that measures the relevance of municipal income from natural resources royalties to conflict relapse due to a non-random allocation of royalties in each municipality. MA is the municipality age to measure institutional development for the municipality "i" in year "t". MI is the municipal investment, and TMI is the total municipality income. MI/TMI is an instrument variable that controls the expected corre-

lation between PMI and MI at the municipality level. All monetary variables are in real terms. Correlations among independent variables show that there is not a high relationship between covariates. However, there is autocorrelation due to unobserved heterogeneity among municipalities that is constant over time (Ci), and there are also temporal effects for each municipality (Di). Uit represents the estimation errors for each municipality in year "t" conditioned to Equation (1).

$$\text{NYPR}_{it} = \beta_{0it} + \beta_1\text{PMI}_{i,t} + \beta_2\text{PTI}_{i,t} + \beta_3\left(\frac{\text{NRR}}{\text{TMI}}\right)_{i,t} + \beta_4\text{MA}_{i,t} + \beta_5 + \left(\frac{\text{MI}}{\text{TMI}}\right)_{i,t} \\ + \beta_6\text{RPD}_{i,t} + \text{C}_i + \text{D}_i + \text{U}_{it} \tag{2}$$

NYPR represents the number of years with the presence of paramilitary organizations the municipality "i" has in year "t". Covariates are the same as those included in Equation (1). In this regard, the number of years with the presence of paramilitary organizations is a function of per capita municipality income (PMI), per capita tax income (PTI), the relative relevance of natural resources royalties on total municipality income (NRR/TMI), municipality age (MA), municipal investment participation as a proportion of total municipality income, rural population density (RPD), unobserved municipality heterogeneity constant over time, and fix temporal effects (Di). Uit is the error estimation of the number of years with paramilitary presence, given Equation (2).

Table 3 compares the research hypothesis between Collier's framework and the identified econometric model. Per capita income measures the cost opportunity of conflict relapse. Higher per capita income brings about a higher opportunity cost, reducing the conflict probability and the presence of paramilitary organizations. Regarding expected gains, the research hypothesis relates higher levels of per capita tax income and higher relative importance in the municipality's total income of natural resources royalties as drivers of conflict relapse and the greater extended presence of paramilitary organizations in each municipality. Those variables replace the share of primary commodities exports used by Collier and Hoeffler at the national level.

**Table 3.** Research hypothesis.

| Cost/Benefit Analysis of Conflict | Collier et al. (2008) | | Research Hypothesis | |
|---|---|---|---|---|
| | **Coefficient** | **Impact** | **Coefficient** | **Expected Impact** |
| Higher levels increase the conflict opportunity cost | Per capita Income | Negative | Per capita municipality income | Negative |
| Higher levels increase expected benefits | Share of primary commodity exports to GDP | Positive | Per capita tax income | Positive |
| | Share of primary commodity exports to GDP | Negative | Relative relevance of natural resources royalties over total municipality income. | Positive |
| Higher levels increase coordination cost/higher levels increase pressure on valuable scare resources (land) | Index of ethnolinguistic fractionalization, ranges 1–100 | Positive | Rural population density | Negative |
| | Index of ethno-linguistic fractionalization2, ranges 1–100 | Negative | | |
| Higher levels increase an incentive to rebellion | Population | Positive | All variables are percapta variables. | |
| Higher levels increase the direct cost of having a private army | | | Relative relevance of municipal investment over total municipality investment. | Negative |
| Older municipalities bring about institutional path dependence | | | Municipality age | Negative |

Note: Based on Collier et al. (2008).

Furthermore, higher rural population density causes higher coordination costs related to a lower probability of conflict relapse and a lower presence of paramilitary organizations in each municipality. The rural population density replaces the index of ethnolinguistic

fractionalization index used by Collier et al. (2008). Moreover, having higher levels of municipality investment as a proportion of total municipality income means better safety networks to protect public investment. Increasing public investment brings about a lower probability of conflict and fewer years with paramilitary organizations in each municipality. Finally, older municipalities have better institutional capabilities to confront paramilitary organizations to reduce the probability of conflict relapse and the number of years with the presence of paramilitary organizations.

### 4.1. Methods and Data

Equations (1) and (2) specify the probability of conflict relapse and the presence of paramilitary organizations in a panel data environment. The dependent variable is a dichotomy variable, meaning the adjusted model involves non-linear panel data. In this regard, Cameron and Trivedi (2009) show how panel data methods can be used to obtain predicted probabilities using logistic and normal distributions. The most relevant assumption in the econometric fitting is related to unobserved heterogeneity (Ci), which caused errors in autocorrelation, meaning that the municipality's differences that were not captured in Equations (1) and (2) could affect the probability of conflict over time. Thus, the random-effects model in the panel data assumes that the unobserved heterogeneity might be modelled using normal distributions of each municipality's unobserved differences through time.

In contrast, the fixed-effects model assumes that the unobserved heterogeny is constant over time. Although there are concerns about the estimator's consistency and efficiency while using non-linear data models to adjust probability, it is shown that whenever the within-group variation in each municipality is lower than the between-group variation of conflict outcomes, the fixed-effects model in the non-linear panel will not be very efficient because such model estimation is mainly based in within variation among municipalities. That is the case for the sample used for adjusting Equations (1) and (2). In addition, Equation (2) uses a tobit model in a panel data environment because not all municipalities show paramilitary organizations' presence. In this regard, some of the municipalities in the sample exhibit zero presence of paramilitary organizations, which means their outcome variable for conflict is censored, given Equation (2).

Data were obtained from the National Historical Center (NHC) and the National Department of Planning (NDP). Data relating to paramilitary organizations about events, number of victims, and years of presence were acquired from the NHC for years 1998, 2001, 2002, 2005, and 2009. In addition, variables for financial records regarding total income, tax income, natural resources royalty income, investment expenditures, rural population, municipal area in squared kilometres, and municipality age were taken from the municipalities database of the NDP. Table 4 describes the variables used in Equations (1) and (2). The model database gathered a total of 5610 observations for 1122 municipalities for each year.

Table 4 exhibits descriptive statistics for dependent variables and identified covariates in Equations (1) and (2). The conflict variable represents paramilitary violence regarding massacre events at the municipality level. The conflict average increased from 1998 to 2009, whereas the standard deviation shows a reduction over the same time. Therefore, the number of municipalities affected by paramilitary violence increased over time, and its average is more representative as the standard deviation suggested more concentration around the mean value. In addition, the variable of the number of years with the presence of paramilitary organizations shows an increasing average from 1998 to 2009, but its standard deviation increases, which shows that paramilitary organizations mobilized around municipalities. Therefore, there were more municipalities affected by paramilitary violence, but the number of years that a paramilitary organization remained in each municipality decreased over time.

**Table 4.** Covariates' descriptive statistics: mean and standard deviation.

| Year | Conflict | NYPR | PMI | PTI | NRR/TMI | MA | MI/TMI | RPD | N |
|------|----------|------|-----|-----|---------|-----|--------|-----|---|
| 1998 | 0.837 (0.36) | 5.01 (6.10) | 0.35 (0.21) | 0.04 (0.06) | 0.02 (0.08) | 152 (110) | 0.36 (0.29) | 37.91 (56.6) | 1122 |
| 2001 | 0.89 (0.30) | 7.11 (6.9) | 0.47 (0.36) | 0.46 (0.06) | 0.047 (0.11) | 152 (110) | 0.28 (0.14) | 37.09 (58.3) | 1122 |
| 2002 | 0.91 (0.28) | 7.94 (7.11) | 0.58 (0.43) | 0.05 (0.06) | 0.049 (0.10) | 152 (110) | 0.30 (0.14) | 37.07 (58.37) | 1122 |
| 2005 | 0.92 (0.25) | 10.67 (7.44) | 0.57 (0.41) | 0.06 (0.07) | 0.035 (0.11) | 152 (110) | 0.34 (0.14) | 37.07 (59.07) | 1122 |
| 2009 | 0.93 (0.24) | 14.5 (7.6) | 0.74 (0.49) | 0.09 (0.11) | 0.048 (0.11) | 152 (110) | 0.36 (0.16) | 37.07 (60.09) | 1122 |
| N | 5610 | 3765 | 5174 | 5166 | 5175 | 5610 | 5175 | 5570 | 5610 |

Per capita municipality income and per capita tax income variables calculated using total amount of municipality income and taxes income and the total population value show an increasing trend from 1998 to 2009. The two variables are unrelated because not all municipalities can collect taxes, and the total income figures are regulated through national regulation of central government transfers to municipalities.

The natural resources royalties, being a central government transference, exhibit higher values in municipalities with more significant natural resource deposits. This raises the potential of a correlation with the error term of Equations (1) and (2), as long as the distribution is not random. To mitigate endogeneity issues and avoid collinearity with municipality per capita income and investment, we constructed the relative share of natural resources royalties income within the total municipal income.

Notably, the relative share of natural resources royalties income within the total income exhibits minimal variation across the data sample over time, with its standard deviation consistently exceeding the mean. This observation underscores the uneven distribution of natural resources across municipalities.

Furthermore, we calculated other fixed factors such as municipality age (MA) and rural population density (RDP) using the municipality's foundation year and total area. Since these municipality characteristics remain constant, the model enhances the efficiency and consistency of coefficient estimation by identifying factors that persist over time, serving as proxies for institutional strength and coordination costs related to potential violent events involving paramilitary organizations.

The data panel remain balanced, encompassing 1122 municipalities each year, resulting in a total of 5610 observations. All variables expressed in monetary values have been deflated using the price consumption index for each respective year.

*4.2. Estimation Results*

4.2.1. Probability of Conflict Occurrence and Impact of Income on Conflict Probability

Table 5 displays the estimated conflict probabilities utilizing four distinct models within a data panel framework. These non-linear probability models, accounting for constant unobserved heterogeneity, employ logistic, probit, fixed-effects, and random-effects models. The estimation also integrates a variable identifying the region to which each municipality belongs. Colombia consists of five regions: (1) the Andean region, (2) the Caribbean region, (3) the Pacific region, (4) the Orinoquia region, and (5) the Amazon region. Among the 5610 total observations, the distribution is as follows: in the Andean region, 3145 observations (56%); in the Caribbean region, 985 observations (17.56%); in the Pacific region, 890 observations (15.86%); and 295 observations (5.26%) each in the Orinoquia and Amazon regions.

**Table 5.** Conflict probability in panel data.

| Pr(Y = Conflict) | Logit Panel | Probit Panel | OLS F.E. | OLS R.E. |
|---|---|---|---|---|
| Municipal income per capita | −0.0928829 *** | −0.0926089 *** | −0.1237967 *** | −0.0941488 *** |
| Municipal tax income per capita | 0.3626739 *** | 0.3181913 *** | 0.3400631 *** | 0.406781 *** |
| % Natural resources royalties in total municipal income | 0.1453259 *** | 0.14652 ** | 0.1723304 *** | 0.1432395 *** |
| Municipality age | −0.0000455 | −0.0000368 | −0.0000442 | −0.0000552 |
| % Municipal investment in total municipal income | −0.10033937 *** | −0.1127963 *** | −0.1375722 *** | −0.1469465 *** |
| Rural population density | −0.0002004 *** | −0.000221 *** | −0.000291 | −0.0003115 *** |
| Andean region | Reference group | Reference group | Reference group | Reference Group |
| Caribbean region | 0.0721543 *** | 0.0694576 *** | 0.0539826 *** | 0.0633122 *** |
| Pacific region | 0.0287513 ** | 0.0249112 ** | 0.0230852 ** | 0.0295414 *** |
| Orinoquia region | 0.0940144 *** | 0.093217 *** | 0.107966 *** | 0.0962675 *** |
| Amazonia region | 0.038487 | 0.0346611 | 0.0257659 | 0.0240842 |
| N | 5166 | 5166 | 5166 | 5166 |
| Number of groups | 5 | 5 | 5 | 5 |
| Min obs. per group | 928 | 928 | 928 | |
| Max obs. per group | 1096 | 1096 | 1096 | |
| Average obs. per group | 1033 | 1033 | 1033 | |
| Wald Chi2 | 201.69 | 237.98 | | 277.56 |
| Prob > Chi2 | 0.0000 | 0.0000 | 0.0000 | 0.0000 |
| Log likelihood | −1203.90 | −1197.65 | | |
| F | | | 31.29 | |
| Prob > F | | | 0.0000 | |
| R2 within | | | 0.0573 | 0.0549 |
| R2 between | | | 0.4484 | 0.1426 |
| R2 overall | | | 0.0487 | 0.0511 |

\* $p < 0.05$; \*\* $p < 0.01$; \*\*\* $p < 0.001$.

The results show that a 1% increase in municipality income per capita leads to a decrease in the probability of conflict across all adjusted models. In the logit, probit, and random-effects models, this increase corresponds to a 9% rise in the probability of a paramilitary conflict while keeping covariates constant at their mean values. The fixed-effects model also exhibits a reduced probability of conflict, but with a more pronounced negative effect, showing a decrease of 12 percentage points in the probability of conflict occurrence. Essentially, greater economic development improves the prospects of reducing conflict recidivism. In simpler terms, a decrease in income per capita heightens the likelihood of conflict occurrence, highlighting the economic underpinnings of conflict. The estimated effect of municipality income per capita is statistically significant at a 1% confidence level in all four adjusted models.

In contrast, municipal tax income per capita exhibits a positive correlation with the probability of paramilitary violence in all the adjusted models. Specifically, a 1% increase in tax income per capita is associated with an escalation in the likelihood of conflict recurrence ranging from 31% to 40%. This effect is statistically significant at a 1% confidence level across all four adjusted models. In theory, a higher level of tax income is expected to be linked to a greater probability of conflict, as it anticipates increased gains from conflict

recurrence. Consequently, municipalities with the capacity to collect taxes are more likely to experience paramilitary violence due to the heightened expected benefits of conflict.

Empirically, as previously noted in the historical review of paramilitary organizations in Colombia, there was a clear incentive to control fiscal resources as a means of financing paramilitary activities through corruption and forced displacement. This was especially evident in municipalities with higher levels of tax income and a weak presence of central government control agencies. It is important to note that only a limited number of municipalities in the sample can collect taxes, which results in increased variability in the sample data.

Similarly, the relative significance of natural resources income within the total municipality income is positively associated with the probability of conflict in all four adjusted models. An increase of 1% in natural resources income corresponds to an increase in conflict probability ranging from 14% to 17%. This effect is statistically significant at a 1% confidence level in the logit, fixed-effects, and random-effects models.

As posited in the research hypothesis, natural resources income serves as a proxy for valuable natural assets within the municipality, representing potential gains from occurrence of violent conflict. Consequently, the higher the economic value of natural resources in proportion to the total municipality income, the greater the likelihood of experiencing paramilitary violence, while holding all other covariates constant at their mean values.

Furthermore, the relative significance of municipal investment, measured in relation to the total municipality income, serves to diminish the probability of paramilitary violence. A 1% increase in municipal investment as a proportion of total income results in a decrease in conflict probability ranging from 10% to 14%. This effect is statistically significant at a confidence level of 1% across all adjusted models. This inverse relationship between municipal investment and conflict probability is associated with the indirect benefits that public investment brings to municipalities. For instance, higher levels of public investment in local infrastructure development, aimed at enhancing economic activities, often entail improved security networks for safeguarding newly established assets at the local level. Consequently, municipalities with greater participation in investment are better equipped to protect their territories from paramilitary violence.

### 4.2.2. Impact of Fixed Effects

The fixed effects related to municipality age and rural population density exert a mitigating influence on the likelihood of conflict occurrence in all the adjusted models. Although these effects are negative in direction, their quantitative impact is nearly negligible. Both variables serve as control measures for the economic factors assessed in the income, tax, and royalties income variables. The negative effect is statistically significant only for rural population density in the logit, probit, and random-effects models. The fixed-effects models exhibit lower explanatory power for both covariates, given that they lack within-municipality variation in the database. Irrespective of the statistical value, the coefficients reflect the institutional strength and coordination costs associated with the occurrence of paramilitary violence. Older municipalities with higher rural population density demonstrate superior institutional capacities for defense and incur greater coordination costs in protecting themselves against paramilitary violence when compared with younger municipalities with lower rural population density.

### 4.2.3. Conflict Occurrence across Regions

Using the Andean region as a reference group due to its higher economic development, the municipalities that belong to the Caribbean region exhibit an increase of between 5% and 7% in paramilitary violence compared with the reference group of municipalities in the Andean region. The estimated effect is statistically significant at a 1% confidence level in all adjusted models. Municipalities located in the Pacific region have an increase of 2% in conflict occurrence compared with the reference group. The estimated effect is statistically

significant at a 5% percent confidence level in the logit, probit, and random effect models. In contrast, it has a statistical significance at a 1% confidence level in the fixed-effects model. Also, municipalities in the Orinoquia region have an increased conflict probability of 2% in all adjusted models. The estimated effect is statistically significant at a 1% confidence level in all the adjusted models. Finally, municipalities in the Amazon region show an increase in conflict probability between 2% and 3% higher than those in the reference group. In this regard, less economic development has a positive impact on conflict occurrence, given that the Andean region reveals higher per capita income (0.56) in the sample compared with the Caribbean region (0.48) and the Pacific region 0.43). Per capita income in the Amazon region (0.56) is equal to the per capita income average in the Andean region, while that in the Orinoquia (0.98) region is higher due to a lower population density compared with the population density in the Andean region.

Table 6 describes the estimated effects of the covariates identified in Equations (1) and (2) on the conflict length using a tobit model in a data panel environment. An increase of 1% in the income per capita brings about, on average, a reduction of 4 years in paramilitary violence, holding all the other covariates constant at their mean value. The estimated effect is statistically at 1% of the level of confidence. Tax income per capita and the relevance of royalties income to total income increase the conflict duration by an average of 14 and 8 years. Thus, a municipality that experiences an increase of 1% in taxes and royalties revenues has a more prolonged presence of paramilitary violence. Positive estimated effects of taxes and royalties revenues are both statistically significant at a 1% percent level of confidence. Investment as a proportion of municipal income also reduces the length of the paramilitary presence in the municipalities that show a higher proportion of investment on average. An increase of 1% in investment as a proportion of the total municipality income reduces the conflict length to 5 years on average. Control variables related to fixed effects of municipality age and rural population density have a negative estimated effect on the number of years with paramilitary presence. Municipalities located in the Caribbean and the Pacific regions have, on average, fewer years with the presence of paramilitary violence than the reference group. In contrast, municipalities in the Amazon and Orinoquia regions have more years of paramilitary presence. All the estimated effects are statistically significant at a 1% confidence level.

**Table 6.** Conflict length tobit panel data.

| Y = Conflict Length in Years | Tobit Panel |
| --- | --- |
| Income per capita | −4.359 *** |
| Tax income per capita | 14.137 *** |
| % Royalties in total income | 8.186 *** |
| Municipality age | −0.001 |
| % Investment in total income | −5.342 *** |
| Rural population density | −0.008 *** |
| Andean region | Reference group |
| Caribbean region | −2.090 *** |
| Pacific region | −1.568 *** |
| Orinoquia region | 1.715 ** |
| Amazonia region | 1.884 ** |
| N | 3.530 |
| Number of groups | 5 |

**Table 6.** *Cont.*

| Y = Conflict Length in Years | Tobit Panel |
| --- | --- |
| Min obs. per group | 634 |
| Max obs. per group | 750 |
| Average obs. per group | 706 |
| Wald Chi2 | 181.71 |
| Prob > Chi2 | 0.0000 |
| Log likelihood | −10,858 |
| Left censored observations | 590 |
| Uncensored observations | 1940 |
| Right censored observations | 0 |

* $p < 0.05$; ** $p < 0.01$; *** $p < 0.001$.

## 5. Conclusions

In conclusion, this paper provides evidence that descriptively contrasts with theoretical assumptions about post-conflict sustainability of peace agreements. From the case study evidence examined in an exploratory manner, it can be stated that economic factors rank prominently among the causes of paramilitary violence in Colombia. Moreover, considering the significance of institutional and security aspects, long-term impacts on economic development through social investment in health and education are crucial, as gaps in economic opportunities become a primary driver of conflict recidivism. In particular, there is a positive effect of higher per capita income in reducing conflict probabilities due to the increased opportunity cost of disrupting economic activities during a conflict. Additionally, a greater reliance on natural resource income, coupled with lower levels of public investment, heightens both the probability and duration of conflicts at the municipal level.

This paper shows a positive correlation between economic income and paramilitary conflict. In agreement with the previous international and national specialized conflict literature, the findings add evidence in favor of the economic causes of civil conflicts. In particular, in the international academic context, Collier (2009) and Collier et al. (2008) identify specific factors associated with the onset of conflicts, such as low per capita income, the state's inability to assert control over its territory, sluggish economic growth, and a high dependency on the export of economic resources. Moreover, at the national level, Giraldo et al. (2015) and Prieto-Bustos and Manrique (2018) concluded that in Colombia, critical investments have to be made in regional economic growth, human capital, social capital, institutional quality, and social development to reduce the probability of conflict recidivism. Although each case requires a contextual analysis, having evidence that supports a theoretical approach to conflict recidivism is a contribution relevant to complementing comparative conflict analysis with a universal theoretical hypothesis for understanding the use of violence for resolving social conflicts.

**Author Contributions:** Conceptualization, W.O.P.B. and J.M.-H.; methodology, W.O.P.B. and J.M.-H.; software program, W.O.P.B. and J.M.-H.; validation, W.O.P.B. and J.M.-H.; formal analysis, W.O.P.B. and J.M.-H.; investigation, W.O.P.B. and J.M.-H.; resources, W.O.P.B. and J.M.-H.; data curation, W.O.P.B. and J.M.-H.; writing—original draft preparation, W.O.P.B. and J.M.-H.; writing—review and editing, W.O.P.B. and J.M.-H.; visualization, W.O.P.B. and J.M.-H.; supervision, W.O.P.B. and J.M.-H.; project administration, W.O.P.B.; funding acquisition, W.O.P.B. and J.M.-H. All authors have read and agreed to the published version of the manuscript.

**Funding:** This research was funded by the Catholic University of Colombia.

**Institutional Review Board Statement:** The study did not require ethical approval.

**Informed Consent Statement:** The study did not involve humans.

**Data Availability Statement:** Research databases are available by request to woprieto@ucatolica.edu.co.

**Conflicts of Interest:** The authors declare no conflict of interest.

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
