# Peer review of "Paramilitary Conflict in Colombia: A Case Study of Economic Causes of Conflict Recidivism"

_socsci, doi:10.3390/socsci13020112_

Round 1

Reviewer 1 Report

Comments and Suggestions for Authors

Thank you for asking me to review this paper. It is an excellent paper and I enjoyed reading it. The findings were interesting and potentially useful in areas beyond Columbia.

In the literature review, an obvious omission is the literature around power elites. The literature of 'Invisible Class Empire' would be of particular benefit to the argument presented by the authors because it would likely address what appears to be an important omission in understanding the influences on the local of powerful players at the national level and why and how paramilitary conflict between local groups is encouraged.

Comments on the Quality of English Language

In spite of the excellent quality of the arguments and the review of literature, paper  needs some substantial work in parts  on the language before it is ready for publication.  It  was seriously unnecessarily difficult to read and needs intensive copy editing. Most  sentences in the paper I had to read multiple times to try to work out what the authors were saying. In almost all cases, it was because the reasoning structure within each sentence was in the opposite order  to that which would make good sense and easy reading. Secondly, on readability, it would be helpful if citations were in past tense rather than present ongoing tense (E.g. Smith 1990 *said* rather that Smith, 1990 says). This would also help resolve many issues relating to prefixes. In a few places, some sentences simply didn't make ANY sense. The paragraphs from line 150 to line 198 are a particularly obvious problem. Along with rewriting to make sense, they would benefit from the addition of subheadings.

Reviewer 2 Report

Comments and Suggestions for Authors

Review: Paramilitary Conflict in Colombia: A case study of economic 2 causes of conflict recidivism

Verdict:

I am in favour of the publication after small revisions

This is a very interesting piece that provides very good ideas on the research topic. The author demonstrates very good empirical knowledge on the area, which is a good foundation for any future publication. Nonetheless, I believe there are some small, minor issues that could be addressed. Therefore, I recommend an acceptance in principle which would require the author to make some small changes to the article in the light of my suggestions below.

Firstly, the paper is coherently and clearly written. The author provides a clear framework for analysis, which is articulated, explained and put in its context within the literature of the article. The author does a good job at outlining some of the main authors to the debate, their contributions and the insights taken from them which are applied in the article. The author provides a good number of important sources and secondary literature in the field. On the whole, the article is well balanced and makes reasonably argued points. I can agree on the larger points the author is making, although I might have some minor quibbles. 

Some general points for changes:

1.              The introduction in its current state could be snappier and clearer: why is this topic interesting? I know the topic is interesting, but this could be underlined more strongly. 

2.              The conclusion of the article should discuss how the findings of the article could be generalised. Can the findings be generalised, if yes, why? If no, why not?

3.              The statistical aspect of the article needs to be checked by a specialist of those methods, which is not the case for me.

Nonetheless, the required changes should not detract from the fact that this is a very clearly argued and articulated article, with a generally convincing thesis, which warrants publication after the revisions have been made.
